# A human centered design approach to define and measure documentation quality using an EHR virtual simulation

**Megha Kalsy**[1,2]*, **Ryan Burant**[1,3], **Sarah Ball**[4], **Anne Pohnert**[4], **Mary A. Dolansky**[1,5]

**1** Frances Payne Bolton School of Nursing, Case Western Reserve University, Cleveland, Ohio, United States of America, **2** Informatics, Decision-Enhancement and Analytic Sciences (IDEAS) Center, George E. Wahlen Veterans Affairs Medical Center, Salt Lake City, Utah, United States of America, **3** Notre Dame College, South Euclid, Ohio, United States of America, **4** MinuteClinic, Woonsocket, Rhode Island, United States of America, **5** Veterans Affairs Northeast Ohio Healthcare System, Cleveland, Ohio, United States of America

☉ These authors contributed equally to this work.

\* megha.kalsy@utah.edu

**Data Availability Statement:** All relevant data are within the manuscript and its Supporting Information files.

## Abstract

Electronic health record (EHR) documentation serves multiple functions, including recording patient health status, enabling interprofessional communication, supporting billing, and providing data to support the quality infrastructure of a Learning Healthcare System. There is no definition and standardized method to assess documentation quality in EHRs. Using a human-centered design (HCD) approach, we define and describe a method to measure documentation quality. Documentation quality was defined as timely, accurate, user-centered, and efficient. Measurement of quality used a virtual simulated standardized patient visit via an EHR vendor platform. By observing and recording documentation efforts, nurse practitioners (NPs) (N = 12) documented the delivery of an Age-Friendly Health System (AFHS) 4Ms (what Matters, Medication, Mentation, and Mobility) clinic visit using a standardized case. Results for timely documentation indicated considerable variability in completion times of documenting the 4Ms. Accuracy varied, as there were many types of episodes of erroneous documentation and extra time in seconds in documenting the 4Ms. The type and frequency of erroneous documentation efforts were related to navigation burden when navigating to different documentation tabs. The evaluated system demonstrated poor usability, with most participants scoring between 60 and 70 on the System Usability Scale (SUS). Efficiency, measured as *click burden* (the number of clicks used to navigate through a software system), revealed significant variability in the number of clicks required, with the NPs averaging approximately 13 clicks above the minimum requirement. The HCD methodology used in this study to assess the documentation quality proved feasible and provided valuable information on the quality of documentation. By assessing the quality of documentation, the gathered data can be leveraged to enhance documentation, optimize user experience, and elevate the quality of data within a Learning Healthcare System.

**Funding:** We are grateful for the support provided by The John A. Hartford Foundation https://www.johnahartford.org/, Author recipient: Mary A. Dolansky, Award Number: SPN00358. Based in New York City, the Foundation is a private, nonpartisan, national philanthropy dedicated to improving the care of older adults. For more than three decades, they have been leaders in the field of aging and health, focusing on creating age-friendly health systems, supporting family caregivers, and improving serious illness and end-of-life care. Their invaluable support has played a crucial role in the success of our project.

**Competing interests:** The authors have declared that no competing interests exist.

# Introduction

Electronic health record (EHR) documentation serves many purposes, such as a (a) record of the patient's health status, (b) way to support coding and billing efforts, (c) platform for inter-professional communication, and (d) source of data to support the quality infrastructure of a Learning Healthcare System (LHS) [1,2]. An LHS is defined by the Agency for Healthcare Research and Quality (AHRQ) as a health system in which internally documented data and processes are purposefully integrated with external evidence in order to provide the knowledge of what needs to be improved [3]. An LHS depends on documentation quality, as the data that are documented are fed back to the managers and healthcare professional staff to evaluate gaps and create cycles to improve care [4,5]. Documentation quality, modeled after the Institute of Medicine's quality of healthcare (IOM, 6 Domains of Healthcare Quality), ensures that documentation is timely, accurate, user-centered, and efficient [6].

Clinically, documentation is key in creating the record of care, which is a legal document describing the clinical interaction between a provider and a patient [7]. This includes documentation of the record of subjective information received through the review of the patient history of present illness, any past medical history, reconciliation of the medication list, any medication or other allergies, and the objective information obtained through physical exam, vital signs, lab work, and other diagnostics. A poorly documented record of care can put an organization at risk for poor quality care and patient experience, patient safety issues, errors, and malpractice liability. In administration, coding and billing rely entirely on an accurate record of documented care, and therefore, billing claims based on poor quality documentation can result in fraud, waste, and abuse. An expanded perspective of quality documentation is in an LHS, where documentation data are essential in the quality infrastructure of improvement. Documented fields in EHRs are used to report trends to healthcare providers and managers in the form of dashboards [8]. A dashboard is a type of health information technology (HIT) that uses data visualization techniques to support clinicians and managers in viewing and exploring data on processes and outcomes of care [9,10]. Dashboards are a key facilitator in quality improvement efforts [11]. If dashboard data are inaccurate, healthcare professionals will lose trust in the system and limit engagement in the use of the dashboards. This in turn limits the healthcare professional's willingness to engage in cycles of improvement as is described in an LHS.

To support an LHS, it is important for healthcare professionals to understand the interface of the quality of EHR documentation and efforts to improve the quality of care [12]. Healthcare professionals currently understand that documentation is an essential record of clinical care and legal purposes, but appreciation for its use to inform gaps in care and to monitor improvement is essential [13]. The User-Technology Interface Design Optimization for Quality Improvement model displays these relationships (Fig 1).

The model includes how quality improvement efforts depend on the User-Technology Interface, Documentation Quality, Dashboard Quality and Health Services Outcomes. It provides a way to comprehensively evaluate the process by operationalizing the dimensions of Documentation Quality including the timeliness, accuracy, user-centeredness, and efficiency. The *Timeliness* of documentation performance refers to the time required to complete documentation tasks. This includes assessing the time taken to complete each tab or screen associated with the EHR documentation and the overall time required to complete the entire set of documentation collectively. *Accuracy* of documentation is defined by the type and frequency of erroneous documentation efforts. Erroneous efforts include timing pauses, mistakes, or omissions that occur during the documentation. Examples of a timing pause is randomly clicking on a screening tab, with uncertainty about where to click precisely. An example of a

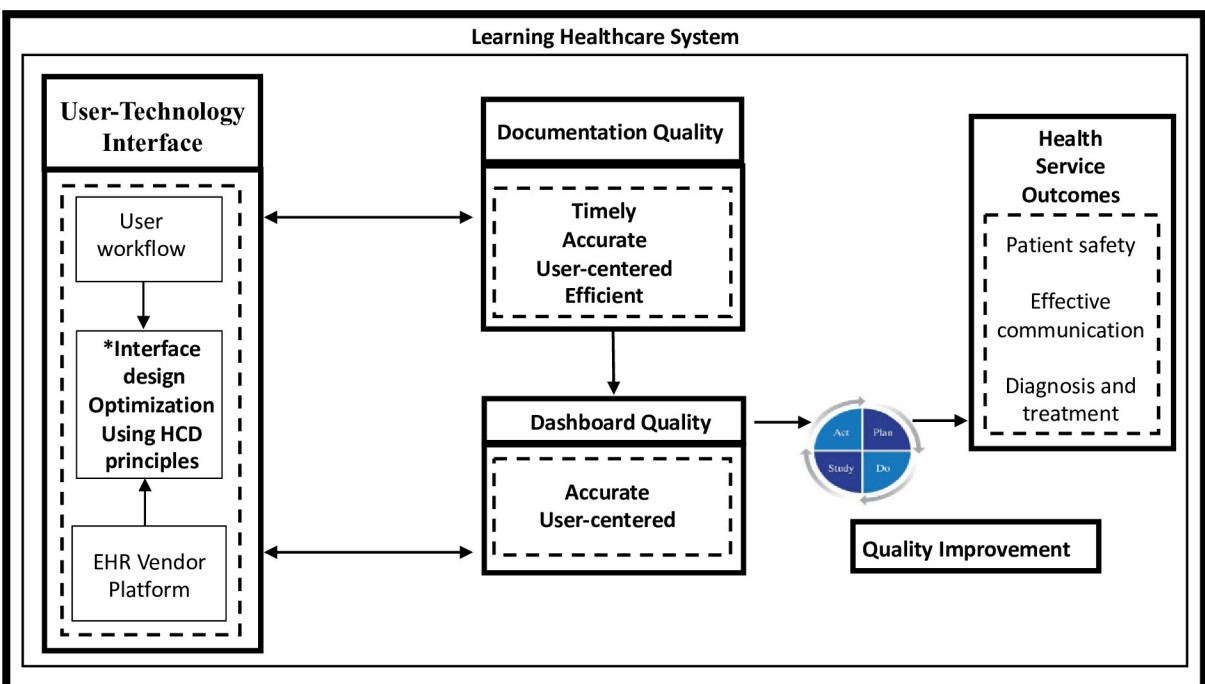

**Fig 1. User-Technology interface design optimization for quality improvement in a learning healthcare system.**

mistake is clicking the wrong tab, or entering incomplete, or incorrect data in the right tab. *User-centeredness* refers to the importance of focusing on the healthcare professional's experience when interacting with the technology interface. User-centered documentation encompasses various factors that contribute to the overall user experience, including ease of use, learnability of, and satisfaction with the EHR interface, and appropriate alignment with the user workflow. It can be assessed using the System Usability Scale (SUS) questionnaire, which provides insights into the participants' perception of the usability of the interface [14,15]. The SUS questionnaire is commonly used to evaluate the usability of informatics application, tools, or software systems, and provides a quantitative measure allowing for comparisons and insights into areas of improvement. *Efficiency* refers to the ease of use of the technology and includes well organized and designed interfaces that support speed of accurate documentation and allow the professional to develop proficiency and competency. Efficiency can be measured by the number of clicks required to complete a task or interact with a system, typically measured per case or per screen [16]. The higher the number of clicks, the higher the *click burden*. Measurement of the click burden provides insights into interaction complexity, steps required to achieve goals, and potential impact on user productivity and engagement.

Documentation quality is influenced by the user-technology interface. The *User-Technology Interface* refers to the interaction between users and healthcare technology, which can occur through various devices, such as computers, tablets, and smartphones. Human-centered design (HCD) or user-centered design (UCD), is a methodology that focuses on designing products, systems, and interfaces to meet the needs, preferences, and abilities of users (Fig 2). In the context of the user-technology interface in healthcare, HCD plays a crucial role [17,18].

HCD involves a series of steps for creating effective solutions. It begins with empathizing with users to understand their needs, followed by defining the problem and ideating potential solutions. Prototypes are then created and tested with users to validate and refine the designs.

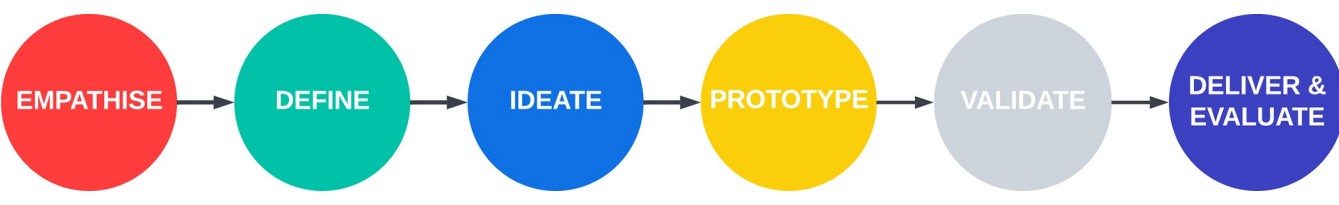

**Fig 2. Human centered design steps.**

The final step is to deliver the solution while considering technical feasibility. Ongoing evaluation ensures that the design meets user expectations and can be iterated upon for continuous improvement. By applying HCD principles, the design of the user-technology interface can prioritize factors such as usability, accessibility, and user satisfaction [17]. In the context of documentation quality, HCD ensures that the user-technology interface design is optimized for timely, accurate, user-centered, and efficient documentation performance (Fig 1).

Documentation quality is necessary for *Dashboard Quality* (e.g. accurate and user-centered). For example, if the data entered are of poor quality (e.g. missing data or inaccurate data), then the dashboard is not accurate. In addition to dashboard accuracy building user trust, dashboard quality is an essential component in quality improvement. In quality improvement, dashboards are used as a reporting mechanism and as a way to monitor the impact of current care delivery improvement efforts. Dashboard quality is essential in the improvement of health system outcomes such as patient safety, quality care, diagnostic and treatment plan excellence, and effective population health and panel management [19].

By addressing the user-technology interface, many threats to documentation quality in healthcare professionals' practices can be addressed. These include unmanageable demands for clinical documentation, inadequate alignment of clinician workflow, and usability issues [20]. For example, when the user-technology interface is optimally designed, it promotes timely, accurate, user-centered, and efficient documentation that results in quality documentation. One example of a non–optimized interface design is the often-performed workaround of the use of free text entry in clinical documents, which appears to be easier for the healthcare professionals. Ensuring that providers have user-friendly structured data entry to avoid free entry errors is key, and would avoid the narrative free-text entry, "copy and paste" or "copy forward" that often leads to documentation errors [21,22].

## Purpose of the study

The purpose of this study is to propose a method to measure the dimensions of documentation quality of the timeliness, accuracy, user-centeredness, and efficiency within a simulated standardized patient visit using an EHR vendor platform. The Age-Friendly Health System (AFHS) is used as an exemplar. The AFHS is an initiative of The John A. Hartford Foundation and the Institute for Healthcare Improvement (IHI), in partnership with the American Hospital Association (AHA) and the Catholic Health Association of the United States (CHA), which began in 2017. AFHS employs a framework called the 4Ms (what Matters, Medication, Mentation, and Mobility) to ensure reliable, evidence-based care for older adults [23,24]. Age-Friendly 4Ms documentation involves recording the assessment and acting on all 4Ms.

Our research aims were as follows:

1. To define documentation quality (timely, accurate, user-centered, and efficient) of healthcare professionals, and

2. To propose a method to measure the responses from healthcare professionals on their documentation quality in the current documentation system.

## Methods

This study employed a descriptive, cross-sectional design. We invited both nurse practitioners (NPs) and Physician Assistants to participate in our study. NPs interested in participating were identified through a snowball sampling process facilitated by regional quality leaders. The sample consisted of 12 NPs from retail health clinics who participated in a simulation consisting of a fictitious patient and a standardized script. Inclusion criteria were successful completion of an AFHS orientation. There were no exclusion criteria.

## Procedures

The study protocol underwent Case Western Reserve University (CWRU) Institutional Review Board (IRB) review and was determined to be exempt. The research team attended AFHS team meetings and reviewed training materials to familiarize themselves with the process of Age-Friendly 4Ms documentation used by NPs and the corresponding locations within the EHR system. An AFHS documentation map (S1 Fig) and Age-Friendly 4Ms data collection tool (S1 Table) were developed based on the reference standard and refined after pilot testing with each NP.

NPs from retail health clinics were identified by regional quality leaders and potential participants were provided an information sheet in an email along with a calendar invite to join a virtual session from the research team. The NPs were recruited in the study between July 25, 2022 –November 15, 2022. Upon entering the virtual EHR setting of the simulated standardized clinic visit, the NPs were read the informed consent form, and a copy was sent via email. The informed consent included the description of the study, benefits, and risks of participating. NPs consented to participation and recording of the session by verbal consent and continuing with the simulation. This consent was witnessed by 3 team members. This informed consent process was approved by the CWRU IRB. Other attendees of the simulation included a standardized patient, the coordinator, and a human-centered design researcher who observed the simulation in both the real-time and recorded sessions. The simulation captured the real-time documentation of the delivery of an AFHS 4Ms clinic visit using a standardized case (S2 Table). Fictitious medications were prepopulated in the EHR vendor platform (S3 Table) and a script was followed by a standardized patient. The simulation provided a standardized case study so that documentation quality could be measured across providers.

Pilot testing was conducted with three NPs, and adjustments were made to the data collection tool based on their feedback. The recording captured the screen documentation of the NPs as they provided care. The recordings were analyzed by two independent researchers who assessed the dimensions of documentation quality (timely, accurate, user-centered, and efficient).

The recordings included capturing the Age-Friendly 4Ms Evaluation Tab, a component within the EHR system that is specifically designed to support the AFHS initiative and provide Age-Friendly care to older adults at the retail health clinic. It is a prebuilt, customizable template that contains 4Ms tools to document and order interventions aligned with the AFHS framework. The Age-Friendly 4Ms Evaluation Tab provides a structured format for documenting assessments, interventions, and care plans related to the 4Ms. The Age-Friendly 4Ms Evaluation Tab is where the Assessments and Actions on the 4Ms is done. The NP is required to navigate to other screening tabs to perform the PHQ2, Mini-Cog, and AGS Beers

Table 1. Documentation quality dimensions and description of measurement.

| Quality Dimension | Description of Measurement | Measured By |
|---|---|---|
| Timely | *Timely* documentation refers to the time required to complete documentation tasks in the EHR. This includes assessing the time taken to complete each tab or screen associated with the EHR documentation and the overall time required to complete the entire set of documentation collectively. | Timely documentation was measured by time (in seconds) to complete EHR documentation (time per tab/screen, and time to complete the AFHS 4Ms Evaluation Tab). |
| Accurate | *Accuracy* is defined by the erroneous documentation, including omission, pauses, and mistakes, committed during the documentation process. Examples of a timing pause is randomly clicking on a screening tab, with uncertainty about where to click precisely. An example of a mistake is clicking the wrong tab, or entering incomplete, or incorrect data in the right tab. | Accuracy was measured in two ways: (1) calculated via extra time in seconds, and (2) type and frequency of the erroneous efforts. |
| User-Centered | *User-centeredness* refers to the importance of focusing on the healthcare professional's experience when interacting with the technology interface. User-centered documentation encompasses various factors that contribute to the overall user experience, including ease of use, learnability of, and satisfaction with the EHR interface, and appropriate alignment with the user workflow. | User-centeredness was assessed using the System Usability Scale (SUS) questionnaire, which provides insights into the participants' perception of the usability when interacting with the Age-Friendly 4Ms Evaluation Tab. |
| Efficient | *Efficiency* refers to the ease of use of the technology, and includes well organized and designed interfaces that support speed of accurate documentation and allow the professional to develop proficiency and competency. | Efficiency was measured by the number of clicks to complete documentation of the Age-Friendly 4Ms (per tab/screen, to complete the Age-Friendly 4Ms Evaluation Tab). |

Medication Synopsis Tab, in order to determine the information to fill into the Age-Friendly 4Ms Evaluation Tab.

A research assistant was trained to populate the data abstracted from the simulation study recordings into the AFHS Data Collection Tool. Data integrity was assessed by the human-factors researcher. The research assistant worked closely with the principal investigator and the human-centered design expert to ensure accurate analysis and interpretation of the data. Finally, the research assistant summarized the data from the 12 AFHS data collection tools for presentation in the results section.

## Measurement

The measurement of *Documentation Quality* is listed in Table 1. *Timely* (time to complete documentation), *accurate* (erroneous documentation included omissions, pauses, and mistakes), *User-Centered* (participants' experience with the Age-Friendly 4Ms Evaluation Tab) and *Efficient* (number of clicks required for documentation) were recorded. Detailed descriptions of these measures can be found in Table 1.

## Additional information obtained from the user

The NPs were asked to complete the SUS questionnaire (S4 Table) to measure the perceived user-centeredness. The SUS included 10 Likert-scale questions assessing usefulness, ease of use, efficiency, learnability, and satisfaction. The psychometric properties of the SUS are sufficiently studied, with reported reliabilities and acceptable levels of convergent validity with other measures of perceived usability and sensitivity. The SUS has demonstrated good internal consistency or reliability. Studies have reported high Cronbach's alpha coefficients for the SUS, typically ranging from 0.95 to 0.97 [25,26]. Another study found the reliability of the SUS was good ($\omega = 0.91$) and total SUS score correlated moderately with the Client Satisfaction Questionnaire (CSQ)-3 (CSQ1 $r_s = .49$, $p < 0.001$; CSQ2 $r_s = .46$, $p < 0.001$; CSQ3 $r_s = .38$, $p < 0.001$), indicating convergent validity [27].

**Table 2. Documentation quality dimension—timeliness measured in seconds.**

| Category (seconds) | Minimum | Maximum | Mean | Median | Standard Deviation |
|---|---|---|---|---|---|
| What Matters | 15 | 70 | 40.5 | 36 | 17.1 |
| Medication | 52 | 648 | 183.7 | 113.5 | 162.9 |
| Mentation | 123 | 478 | 336.2 | 327 | 88.4 |
| Mobility | 15 | 134 | 69.2 | 64.5 | 44.8 |
| Entire 4Ms | 443 | 1085 | 694.1 | 680 | 182.3 |

We included an additional field in our survey to capture the qualitative response from the NPs about navigation, accessing information in the EHR, and screen layout and design. In Question 11 in the survey, we asked, "Is there something that could be changed or added that would enhance the Age-Friendly 4Ms Screening Tab?"

The data were analyzed using descriptive statistics including measures of central tendency (e.g. median, mean) and dispersion (range and standard deviation). Narrative responses were summarized thematically to capture NPs' perspectives qualitatively.

## Results

Descriptive statistics were used to summarize the results for the Documentation Quality dimension of the *Timeliness* (see Table 2), including the documentation completion time for the 4Ms (what Matters, Medication, Mentation, Mobility), and the entire 4Ms documentation. The analysis indicated that there was considerable variability in completion times across different categories. "Mentation" documentation took the most time to complete, with an average of 336.2 seconds (range: 123–478 seconds), while "What Matters" documentation took the least amount of time, with an average of 40.5 seconds (range: 15–70 seconds).

*Accuracy* was measured via erroneous time in seconds (see Table 3) and type and frequency of erroneous documentation efforts (See Table 4). Erroneous documentation efforts in What Matters had a low mean time of 0.82 seconds with a range of 0–9 seconds, indicating minimal erroneous efforts in this category, whereas documentation for "Mobility" had a higher mean erroneous time of 7 seconds, with a range of 0–74 seconds, suggesting that the erroneous efforts in "Mobility" took up significantly more time compared to the other of the 4Ms. We removed an outlier, a NP, from the data for the What Matters category to eliminate its significant skewing effect on the error time in accuracy. This step ensured the accuracy and reliability of our analysis. The types of erroneous documentation efforts were related to navigation to narrative/symptom finder (83.33%), navigation to vital signs (75%), navigation to medication

**Table 3. Documentation quality dimension–accuracy measured in seconds.**

| Documentation Category (seconds) | Minimum | Maximum | Mean | Median | Standard Deviation |
|---|---|---|---|---|---|
| Erroneous time during Medication | 0 | 12 | 1.09 | 0 | 3.62 |
| Erroneous time during Mentation | 0 | 12 | 1.91 | 0 | 4.30 |
| Erroneous time during What Matters | 0 | 9 | 0.82 | 0 | 2.71 |
| Erroneous time during Mobility | 0 | 74 | 7.00 | 0 | 22.24 |
| Total 4Ms Erroneous time | 0 | 74 | 10.82 | 0 | 22.82 |
| Erroneous time during entire visit excluding 4Ms documentation | 3 | 102 | 42.73 | 32 | 36.77 |
| Erroneous time during entire visit including 4Ms documentation | 3 | 117 | 53.55 | 32 | 46.38 |

Note: erroneous documentation = time expended in pauses and mistakes such as going to a wrong field.

**Table 4. Documentation quality dimension–accuracy as measured by type and frequency of erroneous documentation.**

| Type of Erroneous Documentation | Frequency % |
|---|---|
| Navigation to narrative/symptom finder | 83.33 |
| Navigation to Vital Signs | 75 |
| Navigation to medication allergy history | 58.33 |
| Navigation to medical history | 58.33 |
| Performed COVID screening | 33.33 |
| Navigation to Influenza vaccine offer | 33.33 |
| AGS Beer's Criteria medication review location error | 16.67 |

Note: Table only includes the themes with percentage values greater than 8.33%.

allergy history (58.33%), navigation to medical history (58.33%), and performed COVID screening (33.33%).

Documentation Quality of User-Centeredness measured by the SUS found scores ranging from a minimum of 20 to a maximum of 100, with a mean score of 60.21. The standard deviation of the SUS scores was 23.12, indicating that the scores were spread out around the mean, and there was some variability in the data. Qualitative responses from the NPs are found in Fig 3.

The results for the Documentation Quality Dimension of *Efficiency* indicate that medication documentation required fewer clicks on average (11.83 clicks) compared to mentation documentation (24.67 clicks), suggesting that the process for documenting "Medication" was more streamlined (Table 5). However, there was greater variability in the number of clicks required for "Medication" documentation. "What Matters" and "Mobility" documentation required the least number of clicks, with 3.08 and 6.00, respectively.

## Discussion

This was the first study to comprehensively define and propose a strategy to measure documentation quality. Other studies that measured quality documentation included only a comparison of structured and unstructured data elements [22]. The use of a virtual standardized case-based simulation to measure the dimensions of quality documentation proved to be feasible. The method of creating a "testing EHR environment" (sandbox) with a standardized simulated case observing users' documentation provided the opportunity to measure data on dimensions of documentation quality and obtain feedback during the simulation and after. The documentation quality dimensions of the timeliness, accuracy, user-centeredness, and efficiency provided insight into how to improve the design and build of the user interface in LHS.

In terms of the timeliness, our findings suggested that there is considerable variability in completion times across different categories and NP sessions. The simulation method with measurement of the timeliness provided insight into how to improve documentation quality. For example, the reduction in "Mentation" documentation time would be a good way to improve documentation quality. This documentation on average took 5–6 minutes, so a reduction in time would make a large impact on the documentation quality dimension of the timeliness. Possible solutions are to have all documentation within one SmartSet or to prepopulate data that are documented in other clinical notes. For example, the PHQ2 depression score for Mentation was documented by a different HCP and can be pulled into the AFHS clinical note template.

---

**Q11. Is there something that could be changed or added that would enhance the Age-Friendly 4Ms Screening Tab?**

**Theme 1 – Streamlining the screening process:** Several Nurse Practitioners (NPs) suggested changes to make the screening process more efficient and less confusing, such as having all the screenings in one area, carrying over PHQ-2 and PHQ-9 results to the Age Friendly tab, and integrating the PHQ-2, PHQ-9, and Mini-Cog assessments into one tab.
- "Yes, I think making the 'clicks' move down when they expand rather than having to scroll all the way back up to click each 'M.' And have the PHQ-2 and PHQ-9 results carry over to the Age Friendly tab."
- "I would like all screenings to be under the Age Friendly screening tab. It is cumbersome to have to switch to other tabs to complete screening."
- "Adding the synopsis tab for medications in that section in the screening tool – so you don't have to click out of the screening. Same as the PHQ2/9 and Mini-Cog – adding the links in those sections. Or even once clicking Mini-Cog – the questions appear below, so we are not clicking in and out of the 4M screenings."

**Theme 2 – Improving access to information:** Some NPs suggested making it easier to access information, such as adding a synopsis tab for medications, having the synopsis tab appear below the 4Ms box, and adding side effects to all Beers-listed medications for easy review.
- "Adding the synopsis tab for medications in that section in the screening tool – so you don't have to click out of the screening. Same as the PHQ2/9 and Mini-Cog – adding the links in those sections. Or even once clicking Mini-Cog – the questions appear below, so we are not clicking in and out of the 4M screenings."
- "The synopsis tab appearing below the 4Ms box for easier viewing and the potential to add side effects to all Beers-listed medications for easy review."

**Theme 3 – Graphical User Interface (GUI) or user-technology interface accessibility – screen layout and design:** Several NPs suggested changes to the screen layout and design to make the system more user-friendly, such as having all the screenings in one area, making the "clicks" move down when expanding, and adding easier access to the synopsis tab for Beers criteria.
- "Yes, I think making the 'clicks' move down when they expand rather than having to scroll all the way back up to click each M. And have the PHQ-2 and PHQ-9 results carry over to the Age Friendly tab."
- "I would like all screenings to be under the Age Friendly screening tab. It is cumbersome to have to switch to other tabs to complete screening."
- "Adding the synopsis tab for medications in that section in the screening tool – so you don't have to click out of the screening. Same as the PHQ2/9 and Mini-Cog – adding the links in those sections. Or even once clicking Mini-Cog – the questions appear below, so we are not clicking in and out of the 4M screenings."
- "Have all screenings in one area to avoid clicking back and forth."
- "Easier access to the synopsis tab for Beers criteria."

**Theme 4 – Adding comments section:** One NP suggested adding a comments section for each tab to facilitate more specific documentation and avoid charting back and forth between progress notes.
- "I would recommend adding a comment section for each tab, so I can comment or enter a more specific documentation instead of having to chart back and forth between the progress notes."

**Theme 5 – Pre-screen check-in:** One NP suggested incorporating pre-screen check-in for patients who schedule online to streamline the screening process.
- "Pre-screen check-in for patients that schedule online."

**Fig 3. Narrative responses from documentation quality on user-centeredness.**

In terms of accuracy, erroneous time (e.g. pauses, looking for other tabs or screens, and confusion) was found to be a common occurrence during documentation, with certain categories of documentation experiencing longer times than others. There were fewer incidents of

**Table 5. Documentation quality dimension–efficiency measured by click burden.**

| Category (clicks) | Minimum | Maximum | Mean | Median | Standard Deviation |
|---|---|---|---|---|---|
| Medication | 2 | 29 | 11.83 | 7.5 | 10.40 |
| Mentation | 19 | 29 | 24.67 | 24 | 2.90 |
| What Matters | 2 | 5 | 3.08 | 3 | 1.00 |
| Mobility | 2 | 12 | 6.00 | 6 | 2.37 |
| Entire 4Ms | 32 | 67 | 45.17 | 40 | 11.13 |

erroneous time in "What Matters" compared to "Mobility," which had a higher mean error time, suggesting that users were confused about how to document it. This highlights the need for user documentation education and practice to reduce erroneous time, particularly for categories with higher mean times. Education to provide users a demonstration on the correct documentation process and providing time to practice may be necessary for providers using the Age-Friendly 4Ms Evaluation Tab, particularly in navigating specific components of the system.

Regarding user-centeredness, the evaluated system demonstrated poor usability, with most participants giving scores in the range of 60 to 70 on the SUS scale (Mean = 60.8). For interpretation of the score, an SUS > 80.3 = Excellent, 68–80.3 = Good, 51–68 = Poor, and < 51 = Fail [28,29]. SUS scores in the literature were higher suggesting their EHR systems achieved a median usability score that met the industry standard of acceptable usability [30,31]. Notably these scores were achieved after EHR redesign efforts were implemented.

The qualitative component of the SUS provided additional insight. NPs expressed interest in improvements that would make the screening process more efficient and user-friendly, improve access to information, and enable more specific documentation. Suggestions to improve screen layout and design were common among NPs. This feedback highlights the importance of HCD, where insights of end-users are sought to inform and guide the design process. Human-centered design involves actively engaging users in the design process and incorporating their feedback to create more user-friendly and effective systems [7].

The study's findings on *Efficiency* measured as click burden showed that completing medication documentation required fewer clicks compared to mentation documentation. "What Matters" and "Mobility" documentation had a similar number of clicks with minimal variation among participants. These findings suggest that changes using human centered design approaches to the documenting of the 4Ms framework need to be considered for an improvement in documentation workflow and a reduction in the number of clicks. Healthcare providers and electronic health record designers should take these factors into consideration when implementing or optimizing the user-technology interface. Streamlining or simplifying the documentation process, such as by identifying and eliminating redundant or unnecessary clicks, could improve documentation quality [32].

In defining and measuring documentation quality, it is important to complete a workflow documentation analysis (S1 Fig) as the initial step in the analytical process. Traditionally, the initial documentation-workflow analysis is often overlooked, with the focus primarily directed towards assessing usability alone. In the case of this organization, workflow analysis did occur in the initial design and build during focus groups. The documentation workflow analysis in the current study formed the basis for the evaluation checklist. It must be noted that at the time of development of the EHR there were limitations on what was possible from the EHR design perspective. This methods paper proposing a standard simulated virtual case study to measure documentation quality focuses on the systematic approach to documentation analysis

by beginning the journey with the analysis of documentation workflow and then proceeding to determine the appropriate measurement for assessing documentation performance. This sequential approach ensures a methodical and effective exploration of the subject matter.

The method proposed, to measure the dimensions of documentation quality, was feasible based on the operationalized variables of timely, accurate, user-centered, and efficient documentation (Table 1). Based on our model, User-Technology Interface Design Optimization for Quality Improvement in a Learning Healthcare System (Fig 1), the information obtained from a virtual simulation documentation quality can be used to make EHR documentation improvements. As indicated in the model, the documentation improvements will potentially result in the improved performance of the system through accurate documentation being provided to inform quality improvement. The definitions and measurement strategy proposed in the current study contributes to the growing body of science to inform the redesign of the EHR interface.

## Limitations

This study had several limitations that should be considered when interpreting the findings. The research was conducted using a convenience sample of individuals who volunteered to participate. This sampling approach may introduce selection bias, limiting the generalizability of the findings, especially when combined with conducting the study in a single clinical setting, further restricting the applicability of the results to a broader healthcare context.

## Future work

There are several avenues for future research based on the findings and limitations of this study. Testing this approach in other healthcare settings is crucial to determine the generalizability of the approach to other medical institutions and settings. Further testing of the standard virtual simulation to measure documentation quality for other quality metrics will help identify any necessary refinements and assess the applicability of the method in other contexts.

Future research could explore additional quality documentation dimensions, such as task completion, screen pathways, gaze fixation, workload, and satisfaction, and factors that contribute to the time, clicks, and incomplete documentation of errors. Investigating these factors will provide a more comprehensive understanding of quality documentation by including other challenges and potential improvements in the documentation process. This approach will shed light on differences in documentation capture and the inefficient EHR documentation workarounds, allowing for a better understanding of documentation practices and potential areas for optimization.

## Conclusion

Limited studies have focused on defining and measuring the dimensions of documentation quality. Our research aims to address this gap by proposing a method to measure these dimensions. While studies exploring the evolution of user-technology interfaces are often conducted in simulated environments, we seek to go beyond this paradigm. We conducted an innovative approach to measure quality documentation using simulation testing to capture the user's interaction with the technology more comprehensively.

## Supporting information

**S1 Fig. AFHS documentation map.**
(TIF)

**S1 Table. AFHS data collection tool.**
(PDF)

**S2 Table. Fictitious case for age-friendly documentation evaluation.**
(PDF)

**S3 Table. Fictitious medications that were prepopulated in the EHR vendor platform.**
(PDF)

**S4 Table. System Usability Scale (SUS) to measure documentation usability.**
(PDF)

## Acknowledgments

We thank the nurse practitioners from the MinuteClinics located within CVS Pharmacy retail health clinic, and the administrative staff from the Frances Payne Bolton School of Nursing at Case Western Reserve University, for assisting us in this work.

### Disclaimer

The views expressed in this article are those of the authors and do not reflect the position or policy of PLOSOne, the Department of Veterans Affairs, or the United States Government.

## Author Contributions

**Conceptualization:** Megha Kalsy.

**Formal analysis:** Megha Kalsy, Ryan Burant.

**Investigation:** Megha Kalsy.

**Methodology:** Megha Kalsy, Sarah Ball, Mary A. Dolansky.

**Project administration:** Megha Kalsy, Sarah Ball.

**Resources:** Megha Kalsy, Sarah Ball.

**Software:** Megha Kalsy.

**Supervision:** Sarah Ball, Mary A. Dolansky.

**Validation:** Megha Kalsy, Sarah Ball, Mary A. Dolansky.

**Visualization:** Megha Kalsy.

**Writing – original draft:** Megha Kalsy, Anne Pohnert, Mary A. Dolansky.

**Writing – review & editing:** Megha Kalsy, Sarah Ball, Anne Pohnert, Mary A. Dolansky.

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
