## [Decision Letter · Decision Letter 0]

26 Jun 2024

PONE-D-24-15723A Human Centered Design Evaluation of Electronic Health Record Documentation Quality: A Case StudyPLOS ONE

Dear Dr. Kalsy,

Thank you for submitting your manuscript to PLOS ONE. After careful consideration, we feel that it has merit but does not fully meet PLOS ONE’s publication criteria as it currently stands. Therefore, we invite you to submit a revised version of the manuscript that addresses the points raised during the review process.

**The article needs a minor revision.**

We look forward to receiving your revised manuscript.

Kind regards,

Shadab Alam, Ph.D.

Academic Editor

PLOS ONE

2. In the ethics statement in the Methods, you have specified that verbal consent was obtained. Please provide additional details regarding how this consent was documented and witnessed, and state whether this was approved by the IRB.

[This work is supported by a grant from The John A. Hartford Foundation to the Case Western Reserve University Frances Payne Bolton School of Nursing.

We thank the nurse practitioners from the MinuteClinics located within CVS Pharmacy retail health clinic, and the administrative staff from the Frances Payne Bolton School of Nursing at Case Western Reserve University for assisting us in this work. ]

 [MAD;

Award Number: SPN00358;

John A. Hartford Foundation;

https://www.johnahartford.org/

No, the sponsors or funders did not play any role in the study design, data collection and analysis, decision to publish, or preparation of the manuscript]

Reviewers' comments:

Reviewer's Responses to Questions

**Comments to the Author**

1. Is the manuscript technically sound, and do the data support the conclusions?

Reviewer #1: Yes

Reviewer #2: Yes

Reviewer #3: Yes

2. Has the statistical analysis been performed appropriately and rigorously? 

Reviewer #1: Yes

Reviewer #2: Yes

Reviewer #3: Yes

3. Have the authors made all data underlying the findings in their manuscript fully available?

Reviewer #1: Yes

Reviewer #2: Yes

Reviewer #3: Yes

4. Is the manuscript presented in an intelligible fashion and written in standard English?

Reviewer #1: Yes

Reviewer #2: Yes

Reviewer #3: Yes

5. Review Comments to the Author

Reviewer #1: This study aims to address this gap by testing a method to measure these dimensions. They conducted an innovative, user-centred, design-focused approach to measure quality documentation using simulation testing to capture the user’s interaction with the technology more comprehensively.

Reviewer #2: 1- The researcher's contributions need to be better clarified

2- Since all the lines in Table 3 are units of measurement in seconds, there is no need to repeat them and just mention them in the title of the first column.

3- The researcher did not cover his research topic by including sufficient figures

4-The researcher did not provide sufficient comparisons to the works presented in this field

5- The determinants presented by the researcher are very simple and require a fuller explanation along with using the results to create a clearer vision

6- Inclusion of additional references published for the years 2023 and 2024

Reviewer #3: The manuscript titled "A Human Centered Design Evaluation of Electronic Health Record Documentation Quality: A Case Study" provides a thorough examination of EHR documentation quality using a human-centered design approach. The study is well-structured and addresses a critical aspect of healthcare documentation, offering valuable insights and potential improvements. Below are detailed comments and suggestions:

Title and Abstract

The title is clear and reflects the study's content well.

The abstract is comprehensive but could be more concise. Consider summarizing the key points more succinctly.

Introduction

The introduction provides a solid background and clearly states the study's objective. It sets the context well for the reader.

Methods

The study design is appropriate, and the inclusion criteria for participants are well-defined.

Data collection methods are detailed and suitable for the research question.

The analysis methods are generally well-described. However, further clarification on specific statistical techniques used could enhance the understanding.

Results

Results are presented clearly with effective use of tables and figures. The findings related to timeliness, accuracy, user-centeredness, and efficiency are well-explained.

There is a good balance of quantitative and qualitative data, providing a comprehensive view of the documentation quality.

Discussion

The discussion interprets the results effectively, relating them to existing literature and highlighting the implications for EHR documentation improvements.

The limitations of the study are acknowledged. The discussion could benefit from more detailed exploration of potential solutions to the identified issues.

Conclusion

The conclusion summarizes the key findings well and emphasizes the importance of the study's contributions.

References

The references are relevant and up-to-date, supporting the study's context and findings.

Language and Clarity

While the manuscript is generally clear, there are a few typographical and grammatical errors. For example:

In the abstract: "The data obtained from the assessment of documentation quality can be used in efforts to improve the documentation, user experience, and the quality of the data used in a Learning Healthcare System, with the goal of improving healthcare." Consider rephrasing for clarity.

In the Methods section: Ensure consistency in the use of terms such as "Age-Friendly 4Ms Screening Tab" and "AFHS 4Ms documentation."

In the Discussion section: "This highlights the need for user documentation education to reduce erroneous time, particularly for categories with higher mean times." The phrase "user documentation education" could be clarified.

Specific Errors

Inconsistent capitalization of "Nurse Practitioners" vs. "nurse practitioners."

Occasional missing articles, e.g., "the" before "timeliness" in some sentences.

Minor punctuation errors, such as missing commas.

Overall Impression

The manuscript makes a significant contribution to the field by providing a method to evaluate EHR documentation quality using a human-centered design approach. The study is well-conducted, and the findings are valuable for improving EHR systems.

Recommendations for Improvement

Abstract: Make it more concise, ensuring it captures the study's essence within the word limit.

Methods: Further clarify the data analysis process.

Discussion: Expand on potential solutions to the issues identified in the EHR documentation process.

6. PLOS authors have the option to publish the peer review history of their article (what does this mean?). If published, this will include your full peer review and any attached files.

Reviewer #1: No

Reviewer #2: No

Reviewer #3: No

---

## [Author Response · Author response to Decision Letter 0]

24 Jul 2024

Itemized Response to Reviewer Comments

Original Manuscript Title: A Human Centered Design Evaluation of Electronic Health Record Documentation Quality: A Case Study

New Manuscript Title: A Human Centered Design Approach to Define and Measure Documentation Quality using an EHR Virtual Simulation 

Ref.: PONE-D-24-15723

GENERAL JOURNAL REQUIREMENTS

Comment Addressed By

1) Please ensure that your manuscript meets PLOS ONE's style requirements, including those for file naming. We have used the PLOS ONE file naming requirements.

2) In the ethics statement in the Methods, you have specified that verbal consent was obtained. Please provide additional details regarding how this consent was documented and witnessed, and state whether this was approved by the IRB. We have provided details about how this consent was documented and witnessed by 3 team members, and that this was approved by the Case Western Reserve University IRB. 

Addressed on Page 10

3) Please remove any funding-related text from the manuscript and let us know how you would like to update your Funding Statement. We have removed the funding-related statements from the manuscript and included them on the first page (second paragraph) of this rebuttal letter. 

4) Please include your full ethics statement in the ‘Methods’ section of your manuscript file. In your statement, please include the full name of the IRB or ethics committee who approved or waived your study, as well as whether or not you obtained informed written or verbal consent. If consent was waived for your study, please include this information in your statement as well. We have included the full name of the IRB who approved our study. We also included information about the informed consent process. 

Addressed on Page 9-10

5) Please review your reference list to ensure that it is complete and correct. We have reviewed the reference list for accuracy.

We have added the following references: 

Gopidasan B, 2022; Feldman J, 2023; Muylle KM, 2023; Bacchi S, 2023.

Addressed in References #7, #30, #31, #32.

SPECIFIC REVIEWER COMMENTS

Comment Addressed By

1) The researcher's contributions need to be better clarified We are assuming that the reviewer means “aim/purpose” of the paper when stating “contribution”? We have changed the title to reflect the purpose listed in the abstract and discussion. The new title is “A Human Centered Design Approach to Define and Measure Documentation Quality using an EHR Virtual Simulation.” Briefly the contribution of the manuscript is to use a human centered design approach to define documentation and propose a method to measure documentation quality (timeliness, accuracy, user-centeredness, and efficiency) using a virtual simulated standardized patient visit via an EHR vendor platform by observing and recording documentation efforts. 

2) Since all the lines in Table 3 are units of measurement in seconds, there is no need to repeat them and just mention them in the title of the first column. Table 3 was edited and the seconds were removed.

Addressed on Page 15

3) The researcher did not cover his research topic by including sufficient figures We are assuming the reviewer is pointing to the HCD embedded in the research. We added narrative and a figure (Fig. 2) to cover this topic. Currently there are 3 figures within this paper.

Addressed on Page 6

4) The researcher did not provide sufficient comparisons to the works presented in this field 

 We added to the discussion additional comparisons to the work in this field. 

Addressed on Page 17-20

5) The determinants presented by the researcher are very simple and require a fuller explanation along with using the results to create a clearer vision 

 We are assuming that the reviewer means dimensions of documentation quality? We added specifics to the description of the measurement in Table 1. 

Addressed on Page 11

6) Inclusion of additional references published for the years 2023 and 2024 We added updated references Gopidasan B, 2022. We included 3 updated references from 2023: Feldman J, 2023; Muylle KM, 2023; Bacchi S, 2023.

Addressed in References #7, #30, #31, #32.

1) The abstract is comprehensive but could be more concise. Consider summarizing the key points more succinctly. We added the purpose statement to the abstract to capture the study’s essence. 

Addressed on Page 2-3

2) Methods

The study design is appropriate, and the inclusion criteria for participants are well-defined.

Data collection methods are detailed and suitable for the research question.

The analysis methods are generally well-described. However, further clarification on specific statistical techniques used could enhance the understanding. 

 We added a section prior to results that summarizes the statistical techniques used.

Addressed on Page 13

3) Discussion

The discussion interprets the results effectively, relating them to existing literature and highlighting the implications for EHR documentation improvements.

The limitations of the study are acknowledged. The discussion could benefit from more detailed exploration of potential solutions to the identified issues. 

 Thank you for the feedback. We added potential solutions to each section in the discussion.

Addressed on Page 17-20

4) Language and Clarity

While the manuscript is generally clear, there are a few typographical and grammatical errors. For example:

In the abstract: "The data obtained from the assessment of documentation quality can be used in efforts to improve the documentation, user experience, and the quality of the data used in a Learning Healthcare System, with the goal of improving healthcare." Consider rephrasing for clarity. 

 Thank you for this edit. We reviewed the manuscript for typographical errors and changed the language in the abstract.

Abstract Addressed on Page 2-3. Typological errors addressed throughout the manuscript.

5) In the Methods section: Ensure consistency in the use of terms such as "Age-Friendly 4Ms Screening Tab" and "AFHS 4Ms documentation." Thank you for this feedback. We corrected line 224 from “AFHS” to Age-friendly to avoid confusion. We also noted inconsistency in our language with “screening tab and evaluation tab” and corrected this. We also included in the purpose statement what Age-Friendly 4Ms documentation is.

Addressed on Page 10

6) In the Discussion section: "This highlights the need for user documentation education to reduce erroneous time, particularly for categories with higher mean times." The phrase "user documentation education" could be clarified. Clarified what is meant by documentation education.

Addressed on Page 19

7) Specific Errors

Inconsistent capitalization of "Nurse Practitioners" vs. "nurse practitioners." 

 Completed.

Addressed throughout the manuscript

8) Occasional missing articles, e.g., "the" before "timeliness" in some sentences. Completed

Addressed throughout the manuscript

9) Minor punctuation errors, such as missing commas. 

 Thank you, reviewed and completed

Addressed throughout the manuscript

10) Recommendations for Improvement 

Abstract: Make it more concise, ensuring it captures the study's essence within the word limit.

 We added the purpose statement to the abstract to capture the study’s essence. 

Addressed on Page 2

11) Methods: Further clarify the data analysis process.

 Thank you for this suggestion. We added a section before results to clarify the data analysis process. Addressed on Page 13

12) Discussion: Expand on potential solutions to the issues identified in the EHR documentation process.

 Addressed on Page 17-20

---

## [Decision Letter · Decision Letter 1]

5 Aug 2024

A Human Centered Design Approach to Define and Measure Documentation Quality using an EHR Virtual Simulation

PONE-D-24-15723R1

Dear Dr. Kalsy,

We’re pleased to inform you that your manuscript has been judged scientifically suitable for publication and will be formally accepted for publication once it meets all outstanding technical requirements.

Kind regards,

Shadab Alam, Ph.D.

Academic Editor

PLOS ONE

Additional Editor Comments (optional):

Reviewers' comments:

Reviewer's Responses to Questions

**Comments to the Author**

1. If the authors have adequately addressed your comments raised in a previous round of review and you feel that this manuscript is now acceptable for publication, you may indicate that here to bypass the “Comments to the Author” section, enter your conflict of interest statement in the “Confidential to Editor” section, and submit your "Accept" recommendation.

Reviewer #1: All comments have been addressed

Reviewer #2: All comments have been addressed

2. Is the manuscript technically sound, and do the data support the conclusions?

Reviewer #1: Yes

Reviewer #2: Yes

3. Has the statistical analysis been performed appropriately and rigorously? 

Reviewer #1: Yes

Reviewer #2: Yes

4. Have the authors made all data underlying the findings in their manuscript fully available?

Reviewer #1: Yes

Reviewer #2: (No Response)

5. Is the manuscript presented in an intelligible fashion and written in standard English?

Reviewer #1: Yes

Reviewer #2: (No Response)

6. Review Comments to the Author

Reviewer #1: A very interesting and clinically relevant study regarding human centered design approach to define and measure the quality of documentation using an EHR Virtual Simulation.

Reviewer #2: (No Response)

7. PLOS authors have the option to publish the peer review history of their article (what does this mean?). If published, this will include your full peer review and any attached files.

Reviewer #1: No

Reviewer #2: No

---

## [Editor Report · Acceptance letter]

9 Aug 2024

PONE-D-24-15723R1 

PLOS ONE

Dear Dr. Kalsy, 

I'm pleased to inform you that your manuscript has been deemed suitable for publication in PLOS ONE. Congratulations! Your manuscript is now being handed over to our production team.

Kind regards, 

on behalf of

Dr. Shadab Alam 

Academic Editor

PLOS ONE